# Unifying and Certifying Top-Quality Planning

**Primary Keywords:** *(4) Theory;*

## Abstract

The growing utilization of planning tools in practical scenarios has sparked an interest in generating multiple high-quality plans. Consequently, a range of computational problems under the general umbrella of top-quality planning were introduced over a short time period, each with its own definition. In this work, we show that the existing definitions can be unified into one, based on a dominance relation. The different computational problems, therefore, simply correspond to different dominance relations. Given the unified definition, we can now certify the top-quality of the solutions, leveraging existing certification of unsolvability and optimality. We show that task transformations found in the existing literature can be employed for the efficient certification of various top-quality planning problems and propose a novel transformation to efficiently certify loopless top-quality planning.

## Introduction

A body of research on generating plans of top quality (Katz et al. 2018; Speck, Mattmüller, and Nebel 2020; Lee, Katz, and Sohrabi 2023) has recently yielded various computational problems, including unordered top-quality planning (Katz, Sohrabi, and Udrea 2020), subset top-quality planning (Katz and Sohrabi 2022), and loopless top-k planning (von Tschammer, Mattmüller, and Speck 2022). Each of these problems is defined in an ad-hoc manner, addressing specific issues, often identified by an application of interest. For instance, unordered top-quality planning addresses the equivalence of plans that perform the same actions in a different order, a property that holds in many applications. Subset top-quality planning handles unnecessary actions on plans, while the loopless setting specifically addresses unnecessary loops on plans. Clearly, these problems cover only a fraction of possible cases, and there may be various yet unspecified computational problems of practical use. As it is impractical to specify each such computational problem separately, we propose to unify the existing definitions under the framework we call **dominance top-quality planning**. Essentially, any relation over plans defines a computational problem in top-quality planning and the definition specifies whether a set of plans constitutes a solution to the problem. The natural next step is to check whether a set of plans is a solution to the problem. We follow the existing work on certificates of unsolvability (Eriksson, Röger, and

Helmert 2017) and of optimality (Mugdan, Christen, and Eriksson 2023) and propose certifying top-quality planning solutions. We show how to certify top-quality for the unified definition, exploiting existing tools that can certify optimality (Mugdan, Christen, and Eriksson 2023). Further, for the computational problems of unordered top-quality planning and subset top-quality planning, the existing in the literature task transformations (Katz, Sohrabi, and Udrea 2020; Katz and Sohrabi 2022) can be used for efficient certification. For loopless top-quality planning no such transformation exists. We close the gap by introducing a new transformation that allows us to efficiently certify solutions for loopless top-quality planning.

## Background

A *planning task* $\Pi = \langle \mathcal{V}, \mathcal{O}, s_0, s_\star, cost \rangle$ in SAS$^+$ formalism (Bäckström and Nebel 1995) consists of a finite set of finite-domain *state variables* $\mathcal{V}$, a finite set of *actions* $\mathcal{O}$, an *initial state* $s_0$, and the *goal* $s_\star$. Each variable $v \in \mathcal{V}$ is associated with a finite domain $\mathcal{D}(v)$ of values. An assignment of $d \in \mathcal{D}(v)$ to $v \in \mathcal{V}$ is denoted by $\langle v, d \rangle$. A *partial assignment* $p$ maps a subset of variables $\mathcal{V}(p) \subseteq \mathcal{V}$ to values in their domains. For a variable $v \in \mathcal{V}$ and a partial assignment $p$, the value of $v$ in $p$ is denoted by $p[v]$ if $v \in \mathcal{V}(p)$ and is *undefined* otherwise. A full assignment $s$ is called a *state*, and the set of all states is denoted by $\mathcal{S}$. State $s$ is *consistent* with a partial assignment $p$ if they agree on all variables in $\mathcal{V}(p)$, denoted by $p \subseteq s$. Each action $o$ in $\mathcal{O}$ is a pair $\langle pre(o), eff(o) \rangle$, where $pre(o)$ and $eff(o)$ are partial assignments called *precondition* and *effect*, respectively. We denote by $prv(o)$ the precondition restricted to the variables not affected by the action. Furthermore, $o$ has an associated non-negative cost denoted by $cost(o) \in \mathbb{R}^{0+}$. An action $o$ is applicable in state $s$ if $pre(o) \subseteq s$. Applying $o$ in $s$ results in a state denoted by $s[\![o]\!]$, where $s[\![o]\!][v] = eff(o)[v]$ for all $v \in \mathcal{V}(eff)$, and $s[\![o]\!][v] = s[v]$ for all other variables. An action sequence $\pi = \langle o_1 \ldots o_n \rangle$ is applicable in state $s$ if there are states $s_1, \ldots, s_{n+1}$ such that $s = s_1$, $o_i$ is applicable in $s_i$ and $s_i[\![o_i]\!] = s_{i+1}$ for $0 \leq i \leq n$. We denote $s_n$ by $s[\![\pi]\!]$. An action sequence with $s_\star \subseteq s_0[\![\pi]\!]$ is called a *plan*. The cost of a plan $\pi$, denoted by $cost(\pi)$ is the sum of the costs of the actions in the plan. The set of all plans is denoted by $\mathcal{P}_\Pi$, and an *optimal* plan is a plan in $\mathcal{P}_\Pi$ with the minimum cost. Next, we present the **top-k** planning problem, as de-

fined by Sohrabi et al. (2016) and Katz et al. (2018). Given a classical planning task $\Pi$ and a natural number $k$, find a set of plans $P \subseteq \mathcal{P}_\Pi$ that satisfy the following properties: (I) For all plans $\pi \in P$, if there exists a plan $\pi' \in \mathcal{P}_\Pi$ such that $cost(\pi') < cost(\pi)$, then $\pi' \in P$, and (II) $|P| \leq k$, and if $|P| < k$, then $P = \mathcal{P}_\Pi$. Note that cost-optimal planning is a special case of top-$k$ planning for $k = 1$. Extending cost-optimal planning, **top-quality** planning (Katz, Sohrabi, and Udrea 2020) deals with finding *all* plans of up to a specified cost. Given a planning task $\Pi$ and a number $q \in \mathbb{R}^{0+}$, find the set of plans $P = \{\pi \in \mathcal{P}_\Pi \mid cost(\pi) \leq q\}$. In some cases, an equivalence between plans can be specified, allowing to possibly skip some plans, if equivalent plans are found. The corresponding problem is called **quotient top-quality** planning and it is formally specified as follows. Given a planning task $\Pi$, an equivalence relation $N$ over its set of plans $\mathcal{P}_\Pi$, and a number $q \in \mathbb{R}^{0+}$, find a set of plans $P \subseteq \mathcal{P}_\Pi$ such that $\bigcup_{\pi \in P} N[\pi]$ is the solution to the top-quality planning problem. The most common case of such an equivalence relation is when the order of actions in a valid plan is not significant from the application perspective. In other words, when you can reorder some of the actions in a plan and still get a valid plan. The corresponding problem is called **unordered top-quality** planning and is formally specified as follows. Given a planning task $\Pi$ and a number $q \in \mathbb{R}^{0+}$, find a set of plans $P \subseteq \mathcal{P}_\Pi$ such that $P$ is a solution to the quotient top-quality planning problem under the equivalence relation $R_U = \{(\pi, \pi') \mid \pi, \pi' \in \mathcal{P}_\Pi, \mathrm{MS}(\pi) = \mathrm{MS}(\pi')\}$, where $\mathrm{MS}(\pi)$ is the multi-set of the actions in $\pi$. Going beyond equivalence relations, let $R_\subset = \{(\pi, \pi') \mid \mathrm{MS}(\pi) \subset \mathrm{MS}(\pi')\}$ denote the relation defined by the subset operation over plan action multi-sets. The **subset top-quality** planning problem (Katz and Sohrabi 2022) is defined as follows. Given a planning task $\Pi$ and a natural number $q$, find a set of plans $P \subseteq \mathcal{P}_\Pi$ s.t. (i) $\forall \pi \in P$, $cost(\pi) \leq q$, and (ii) $\forall \pi' \in \mathcal{P}_\Pi \setminus P$ with $cost(\pi') \leq q$, $\exists \pi \in P$ s.t. $(\pi, \pi') \in R_\subset$. In words, a plan $\pi$ with $cost(\pi) \leq q$ may be excluded from the solution to the subset top-quality planning problem only if its subset is part of the solution. Note, while a top-quality and unordered top-quality solutions are also subset top-quality solutions, we are interested in finding the smallest (in the number of plans) such solutions. While unordered top-quality solutions can be of infinite size, the smallest subset top-quality planning solutions are always finite. Yet another problem is the **loopless top-k** planning problem (von Tschammer, Mattmüller, and Speck 2022). It is defined in the literature similarly to the top-k planning problem, where the set of all plans $\mathcal{P}_\Pi$ is replaced with the set of all loopless plans $\mathcal{P}_\Pi^{\ell\ell}$. Here, we consider the corresponding **loopless top-quality** planning problem. We therefore define the top-quality variant as follows. Given a planning task $\Pi$ and a number $q \in \mathbb{R}^{0+}$, find the set of plans $P = \{\pi \in \mathcal{P}_\Pi^{\ell\ell} \mid cost(\pi) \leq q\}$.

## Unifying Top-quality Planning

We start by noticing that the definition for quotient top-quality planning can be rewritten similarly to the one for subset top-quality planning, by requiring the plans of bounded by $q$ costs missing from the solution to be in relation to the ones that are in the solution. With that, given

any relation $R$, we extend the Definition 1 of Katz and Sohrabi (2022) as follows.

**Definition 1** *Let $\Pi$ be some planning task, $\mathcal{P}_\Pi$ be the set of its plans, and $R$ be some relation over $\mathcal{P}_\Pi$. The **dominance top-quality planning** problem is defined as follows. Given a natural number $q$, find a set of plans $P \subseteq \mathcal{P}_\Pi$ such that*

1. $\forall \pi \in P$, $cost(\pi) \leq q$,
2. $\forall \pi' \notin P$ with $cost(\pi') \leq q$, $\exists \pi \in P$ such that $(\pi, \pi') \in R$,
3. $P$ is minimal under $\subseteq$ among all $P' \subseteq \mathcal{P}_\Pi$ for which conditions 1 and 2 hold.

Observe that for an empty relation we get the top-quality planning problem, for $R_U$ we get the unordered top-quality planning (Katz, Sohrabi, and Udrea 2020), while for $R_\subset$ we get sub-multi-set top-quality planning (Katz and Sohrabi 2022). It is worth noting that in these publications, the third condition was not part of the definition, but the aim was to find solutions of minimal size.

Moving on to loopless top-quality planning, while the existing definition is not expressed via a relation over plans, there can be various such relations. While varying in detail, the idea is the same – a plan without loops dominates plans with loops. Concrete relations may require that the loopless plan could be obtained from the one with loops by removing these loops. A relation we adopt in this work, however, is somewhat more general: $(\pi, \pi') \in R_{\ell\ell}$ if and only if (a) $\pi$ is a loopless plan and (b) if $S'$ are the states traversed by $\pi$, then $\pi'$ traverses some $s \in S'$ more than once.

**Theorem 1** *The dominance top-quality planning problem for $R_{\ell\ell}$ is the loopless top-quality planning problem.*

The proof is deferred to the supplementary material.

While none of the common properties of relations (reflexivity, transitivity, and symmetry) are required to make Definition 1 well defined, it is worth discussing the relations we considered so far. The quotient top-quality planning problem in general and unordered top-quality in particular deal with equivalence relations. On the other hand, the relations $R_\subset$ and $R_{\ell\ell}$ are transitive and anti-symmetric, but they are not reflexive and therefore not a partial order. In principle extending these relations by adding the elements $(\pi, \pi)$ for all $\pi \in \mathcal{P}_\Pi$ would not affect what is considered to be a solution under the definition. In general, if $(\pi, \pi') \in R$ and $(\pi', \pi'') \in R$, if both $\pi'$ and $\pi''$ are not in $P$, while $\pi$ is, transitivity would suffice to ensure condition 2. It is not strictly necessary though. An example is $R = \{(\pi_a, \pi_b), (\pi_c, \pi_d), (\pi_b, \pi_d), (\pi_d, \pi_b)\}$ over the set of plans $\mathcal{P}_\Pi = \{\pi_a, \pi_b, \pi_c, \pi_d\}$, all of the same cost $q$. The set $P = \{\pi_a, \pi_c\}$ is a solution to the $R$ dominance top-quality problem according to Definition 1. Note that the example relation is neither symmetric nor anti-symmetric.

## Certificate of Top Quality

Given a planning task $\Pi$ and a set of plans $P \subseteq \mathcal{P}_\Pi$, note that it is possible to obtain another (transformed) planning task $\Pi_P$ with the set of plans being $\mathcal{P}_\Pi \setminus P$ (Katz et al. 2018). The family of such transformations is generally called *plan forbidding*, and several such transformations exist in the literature, forbidding plans and their reorderings (Katz and

Sohrabi 2020; Katz, Sohrabi, and Udrea 2020) or plans and their super-multi-sets (Katz and Sohrabi 2022).

We start with certifying the top-quality planning problem.

**Theorem 2** *A set of plans $P$ can be checked for being a solution to the top-quality planning problem.*

**Proof:** $P$ is a solution to the top-quality planning problem if and only if (a) $\forall \pi \in P$, $cost(\pi) \leq q$, and (b) $\Pi_P$ has no plans of cost smaller or equal to $q$. That means that we can certify top-quality planning solutions for $\Pi$ by certifying optimality for the transformed task $\Pi_P$ (Mugdan, Christen, and Eriksson 2023). $\square$

Moving now to dominance top-quality planning, for a set of plans $P$, we define an *extended under $R$* set of plans

$$\overline{P} := P \cup \bigcup_{\pi \in P} \{\pi' \in \mathcal{P}_\Pi \mid cost(\pi') \leq q, (\pi, \pi') \in R\}.$$

**Theorem 3** *If $P$ is a solution to the dominance top-quality planning problem, then $\overline{P}$ is a solution to the top-quality planning problem.*

The proof is technical, deferred to the supplementary material. We can now certify solutions to the dominance top-quality planning problem.

**Theorem 4** *A set of plans $P$ can be checked for being a solution to the dominance top-quality planning problem.*

**Proof:** We can certify $P$ in two steps.

- Certify $\overline{P}$ to be a solution to the top-quality planning problem, and

- For every $\pi \in P$, show that $\overline{P \setminus \{\pi\}}$ is not a solution to the top-quality planning problem.

The latter is needed for certifying the condition 3 of Definition 1. The first two conditions of Definition 1 imply that if $P$ is a solution, then so is its superset. Therefore, it is sufficient to ensure that removing any single plan prevents the extended set from being a top-quality solution. $\square$

Given $P$, it can be impossible to compute $\overline{P}$ explicitly. One example is the $R_{\ell\ell}$ relation, where while the set of all loopless plans $\mathcal{P}_\Pi^{\ell\ell}$ is finite and therefore all solutions to the loopless top-quality planning problem are finite, the extended under $R_{\ell\ell}$ set can be infinite. In what follows, we show how to efficiently certify dominance top-quality without explicitly computing $\overline{P}$.

### Efficient Computation

In practice, when $R$ is specified implicitly, like $R_U$ for *unordered top-quality*, $R_\subset$ for *subset top-quality* or $R_{\ell\ell}$ for *loopless top-quality*, it may be computationally intensive or even infeasible to explicitly generate the extended set $\overline{P}$ from a solution $P$. In such cases, instead of transforming $\Pi$ into a planning task that forbids the set of plans $P$, we can transform into a planning task $\overline{\Pi}_P$ that forbids the entire extended set $\overline{P}$. Such transformations exist for the cases of $R_U$ (Katz and Sohrabi 2020) and $R_\subset$ (Katz and Sohrabi 2022). Similarly to the case of top-quality planning, we can now

certify $P$ to be a solution to the unordered respectively subset top-quality planning by certifying optimality of the respective transformation. Such transformation however does not exist for loop-less top-quality planning. In what follows, we present a novel transformation for the $R_{\ell\ell}$ relation, allowing us to certify $P$ to be a solution to the loopless top-quality planning.

**Forbidding Plans With Loops** Here we present a novel transformation for forbidding plans with loops. Specifically, for a plan $\pi$, it forbids $\pi$ and all $\pi'$ such that $(\pi, \pi') \in R_{\ell\ell}$.

The idea is based on the transformation that forbids a single plan (Definition 3 by (Katz et al. 2018)), with an additional dimension added, ensuring that no action enters the already traversed states of $\pi$. Each original action is extended with additional preconditions and effects, allowing to capture whether the execution (a) is following the prefix of the input plan $\pi$, (b) diverging from $\pi$, or, (c) has already diverged from $\pi$. Each of these is further split to sub-cases. Figure 1 visualizes the various sub-cases in different colors. It shows three different areas of the state space, with the bottom one including the initial state, the top one including only sink states, and the middle one, where all the goal states are. In what follows, we will refer to the colors of the edges to simplify the explanation. Further, in what follows, for simplicity, we use disjunctive preconditions, which can be compiled away by introducing multiple action instances.

Let $\Pi = \langle \mathcal{V}, \mathcal{O}, s_0, s_\star, cost \rangle$ be a planning task and $\pi = o_1 \cdot \ldots \cdot o_n$ be some plan for $\Pi$ traversing the states $s_0, \ldots, s_n$. For every $0 \leq i \leq n$, let

$$\mathcal{O}_i := \{o \in \mathcal{O} \mid prv(o) \cup eff(o) \subseteq s_i\}$$

denote the actions that may lead to the state $s_i$. For an action $o \in \mathcal{O}_i$, let

$$cond_i(o) := s_i \setminus (prv(o) \cup eff(o))$$

denote the extra condition that would ensure the state achieved by applying $o$ is exactly $s_i$. We denote the negation of $cond_i(o)$ by

$$\overline{cond_i(o)} := \bigvee_{v \in \mathcal{V}(cond_i(o))} \mathcal{D}(v) \setminus \{cond_i(o)[v]\}.$$

The task $\Pi_\pi^{\ell\ell} = \langle \mathcal{V}', \mathcal{O}', s_0', s_\star', cost' \rangle$ is defined as follows.

- $\mathcal{V}' = \mathcal{V} \cup \{\overline{v}^d, \overline{v}^e\} \cup \{\overline{v}_i \mid 0 \leq i \leq n\}$, with $\overline{v}^d$ and $\overline{v}^e$ being binary variables and $\overline{v}_i$ being ternary variables,

- $s_0'[v] = s_0[v]$ for all $v \in \mathcal{V}$, $s_0'[\overline{v}^d] = F$, $s_0'[\overline{v}^e] = F$, $s_0'[\overline{v}_0] = C$, and $s_0'[\overline{v}_i] = 0$ for all $i > 0$, and

- $s_\star'[v] = s_\star[v]$ for all $v \in \mathcal{V}$ s.t. $s_\star[v]$ defined, $s_\star'[\overline{v}^d] = T$, and $s_\star'[\overline{v}^e] = F$.

As mentioned before, the actions $\mathcal{O}'$ extend the actions in $\mathcal{O}$ by introducing additional preconditions and effects, with multiple copies covering the various cases. Specifically,

$$o_i^f = \langle pre(o_i) \cup \{\langle \overline{v}^e, F \rangle, \langle \overline{v}^d, F \rangle, \langle \overline{v}_{i-1}, C \rangle\},$$
$$eff(o_i) \cup \{\langle \overline{v}_{i-1}, 1 \rangle, \langle \overline{v}_i, C \rangle\} \rangle$$

are the copies of the actions on $\pi$ (colored black in Figure 1) that follow the execution of the plan $\pi$, until a diverging from

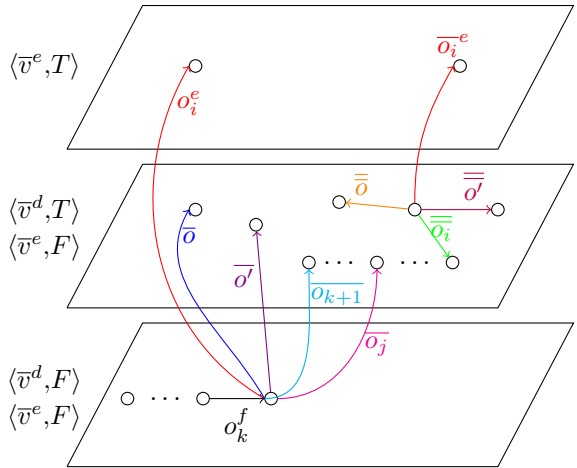

Figure 1: Visualization of the transformation.

$\pi$ is happening. When that happens, there are five options. First, the first action $o \neq o_{k+1}$ diverging from $\pi$ (after $op_1 \cdot \ldots \cdot o_k$ were applied) is in $\mathcal{O}_i$ for some $i < k$ and it reaches an already traversed state $s_i$, with an action copy (red)

$$o_i^e = \langle pre(o) \cup \{\langle \overline{v}^e, F \rangle, \langle \overline{v}^d, F \rangle, \langle \overline{v}_i, 1 \rangle, \langle \overline{v}_k, C \rangle\} \cup cond_i(o),$$
$$eff(o) \cup \{\langle \overline{v}^e, T \rangle\}\rangle.$$

Second, it reaches a not yet traversed state $s_j$ for $j > k+1$ with an action copy (magenta)

$$\overline{o_j} = \langle pre(o) \cup \{\langle \overline{v}^e, F \rangle, \langle \overline{v}^d, F \rangle, \langle \overline{v}_j, 0 \rangle, \langle \overline{v}_k, C \rangle\} \cup cond_j(o),$$
$$eff(o) \cup \{\langle \overline{v}^d, T \rangle, \langle \overline{v}_j, 1 \rangle, \langle \overline{v}_k, 1 \rangle\}\rangle.$$

Third, it reaches the state $s_{k+1}$ with an action copy (cyan)

$$\overline{o_{k+1}} = \langle pre(o) \cup \{\langle \overline{v}^e, F \rangle, \langle \overline{v}^d, F \rangle, \langle \overline{v}_k, C \rangle\} \cup cond_{k+1}(o),$$
$$eff(o) \cup \{\langle \overline{v}^d, T \rangle, \langle \overline{v}_k, 1 \rangle, \langle \overline{v}_{k+1}, 1 \rangle\}\rangle.$$

Forth, it reaches a different state with an action copy (blue)

$$\overline{o} = \langle pre(o) \cup \{\langle \overline{v}^e, F \rangle, \langle \overline{v}^d, F \rangle, \langle \overline{v}_k, C \rangle\} \cup \bigwedge_{i=0}^{k-1} \overline{cond_i(o)},$$
$$eff(o) \cup \{\langle \overline{v}^d, T \rangle\} \cup \{\langle \overline{v}_k, 1 \rangle\}\rangle.$$

Fifth and last, if the action $o$ is not in any $\mathcal{O}_i$, $0 \leq i \leq n$, it reaches a different state with an action copy (violet)

$$\overline{o'} = \langle pre(o) \cup \{\langle \overline{v}^e, F \rangle, \langle \overline{v}^d, F \rangle, \langle \overline{v}_k, C \rangle\},$$
$$eff(o) \cup \{\langle \overline{v}^d, T \rangle\} \cup \{\langle \overline{v}_k, 1 \rangle\}\rangle.$$

Once diverged from $\pi$, the action copies only need to record whether they reached one of the states $s_i$ more than once. This can be done with the following cases:

$$\overline{\overline{o_i}} = \langle pre(o) \cup \{\langle \overline{v}^e, F \rangle, \langle \overline{v}^d, T \rangle, \langle \overline{v}_i, 0 \rangle\} \cup cond_i(o),$$
$$eff(o) \cup \{\langle \overline{v}_i, 1 \rangle\}\rangle,$$

for reaching $s_i$ for the first time (green),

$$\overline{o_i}^e = \langle pre(o) \cup \{\langle \overline{v}^e, F \rangle, \langle \overline{v}^d, T \rangle, \langle \overline{v}_i, 1 \rangle\} \cup cond_i(o),$$
$$eff(o) \cup \{\langle \overline{v}^e, T \rangle\}\rangle,$$

for reaching $s_i$ for the second time (red), or

$$\overline{\overline{o}} = \langle pre(o) \cup \{\langle \overline{v}^e, F \rangle, \langle \overline{v}^d, T \rangle\} \cup \bigwedge_{i=0}^{k-1} \overline{cond_i(o)}, \ eff(o)\rangle,$$

(colored orange) for reaching a state not on $\pi$. Here as well, if the action $o$ is not in any $\mathcal{O}_i$, $0 \leq i \leq n$, it reaches a different state with an action copy (purple)

$$\overline{\overline{o'}} = \langle pre(o) \cup \{\langle \overline{v}^e, F \rangle, \langle \overline{v}^d, T \rangle\}, \ eff(o)\rangle.$$

The costs of these copies is the same as the cost of the original action. We say that $\Pi_\pi^{\ell\ell}$ is the loopless transformation of $\Pi$ under $\pi$.

**Theorem 5** *Let $\Pi$ be a planning task, $\pi$ be its plan and $\Pi_\pi^{\ell\ell}$ be the loopless transformation under $\pi$. Then $\Pi_\pi^{\ell\ell}$ forbids exactly $\pi$ and all plans dominated by $\pi$.*

The proof is deferred to the supplementary material.

## Certificate of Top-k

For top-k planning problem as well, we can certify $P$ to be top-k solution.

**Theorem 6** *A set of plans $P$ can be checked for being a solution to the top-k planning problem.*

**Proof:** First, if $|P| < k$, we certify unsolvability of $\Pi_P$. Otherwise, let $m = \max_{\pi \in P} cost(\pi)$ and $P_1 := \{\pi \in P \mid cost(\pi) < m\}$ and $P_2 := \{\pi \in P \mid cost(\pi) = m\}$ be the partition of $P$ into the set of plans of maximal cost and the set of plans of lower than maximal cost. If $P_1$ is empty, then all plans in $P$ are of the same cost $m$, and we can certify top-k planning by certifying optimality of m. Now, assume that $P_1$ is not empty and let $q = \max_{\pi \in P_1} cost(\pi)$ be the maximal cost in $P_1$. Observing that $P_1$ must be a solution for the top-quality planning problem for $q$, we can certify top-k planning solution in two steps: (I) certify $P_1$ to be a solution to the top-quality planning problem, and (II) certify optimality of $m$ for $\Pi_{P_1}$. $\square$

## Discussion and Future Work

We present a definition that unifies existing computational problems under the umbrella of top-quality planning. Based on the definition, we show that certification of top-quality can be obtained with the help of task transformations and certification of optimality. We further show that existing task transformations can be used for efficient certification of unordered and subset top-quality planning problems. Finally, we present a novel transformation and use it for certifying loopless top-quality planning.

In the future, we would like to certify planning algorithms for the various top-quality problems. While the ForbidIterative planners are relatively straightforward to certify, other methods, such as $K^*$ search or symbolic search are more challenging to certify. Same can be said for search pruning techniques used in top-quality planners, such as symmetry reduction or partial order reduction. Another interesting direction for future work is to use the suggested in this work transformation for loopless top-quality planning, e.g., as part of the ForbidIterative framework. While the transformation significantly increases the number of actions, it might still pay off in several domains.

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
