# OpenReview forum: "Unifying and Certifying Top-Quality Planning"
_icaps-conference.org/ICAPS/2024/Conference — ICAPS 2024_

### Official Review · Reviewer_dtum · 2024-01-20

**Significance And Importance:** 2
**Soundness:** 3
**Novelty:** 2
**Clarity:** 3
**Overall Evaluation:** 1
**Confidence:** 4

**Weaknesses:**

1: Minor weaknesses that are easily fixable.

**Contributions Of The Paper:**

The paper addresses top-quality planning. First it introduces a definition that unifies existing variants of this computational problem. Then it shows that (1) certification of top-quality can be obtained with the help of task transformations and certification of optimality, (2) existing task transformations can be used for "efficient" certification of un-ordered and subset top-quality planning problems. Finally, it presents a novel transformation for certifying loopless top-quality planning.

**Ethical Considerations:**

(5) Excellent: The paper comprehensively addresses all of the applicable ethical considerations

**Nomination For Best Paper:**

No

**Questions For Authors:**

Can you clarify why "instead of transforming \Pi into a planning task that forbids the set of plans P, we can transform into a planning task \oveline{\Pi}_P that forbids the entire extended set \overline{P}" is more efficient?
What  does "efficient computation" mean?
What is the computational cost of the transformation proposed for forbidding plans with loops?

**Reproducibility:**

0: N/A - nothing to reproduce.

**Strengths Of The Paper:**

The results in the paper help to better characterize and understand the problem of computing top-quality plans, considering the variants of this problem previously investigated. It also proposes techniques for certifying top-quality planning solutions.

**Weaknesses Of The Paper:**

Readability could be improved by adding some more content to the background section (although I understand this is not easy for a short paper). E.g., describing the problem of certifying top-quality planning solutions, which is quite relevant for this paper, and indicating the complexity of the considered computational problems.
The paper uses the expression "efficient computation" in an inform way. I would have liked to see a more detailed/formal analysis about efficiency (either theoretical or experimental/practical) and/or a more elaborated explanation of what efficiency means in this context.

---

> ### Author Rebuttal · Authors · 2024-01-27
>
> We thank the reviewer for the feedback.
>
> * That is indeed a poorly composed sentence. The intention was that instead of first generating \overline[P} from P and then transforming the task based on Π and \overline[P} as input, we can directly transform Π into a task that forbids the entire set \overline[P} based on Π and P as input, with a dedicated transformation per unordered/subset/loopless top-quality planning problems.  We will modify and clarify in the final version.
>
> * It is poly-time in the input size to transform the planning task for the existing and the newly proposed transformations. Please see the response to Q3 of reviewer KFqW for additional details on the size of the transformations.

---

### Official Review · Reviewer_KFqW · 2024-01-21

**Significance And Importance:** 2
**Soundness:** 2
**Novelty:** 2
**Clarity:** 2
**Confidence:** 3

**Weaknesses:**

0: Minor weaknesses requiring some work to be addressed for the paper to be accepted.

**Contributions Of The Paper:**

The authors describe several variants of Top-K planning, which is the problem of finding a set of K 'best' plans.  The authors unify these variants using a general purpose 'dominance' relationship combined with notions of equivalence classes among plans, so that only one of each equivalence class is necessary in the solution.  The authors use 'forbidding planning' to describe various certification results for Top-K, and describe a new forbidding relationship for loopless planning.

**Ethical Considerations:**

(1) Not Applicable: The paper does not have any ethical considerations to address

**Nomination For Best Paper:**

No

**Overall Evaluation:**

-1: (weak reject)

**Questions For Authors:**

Specific comments:

p. 1: I see different notational conventions used for non-negative real costs: $\mathbb{R}_{geq 0}$ and $\mathbb{R}_+$ . I don’t think I’ve ever seen $\mathbb{R}^{0+}$.  (A nit)
p. 2: if plans can be partial order, we wouldn’t need unordered top quality, right?  (Top-K would be sufficient). (This is the only version of top-k I find trivially objectionable, all the others are very reasonable.)
p. 2: I don’t understand why unordered top quality plan solutions can be of infinite size. Is it because action costs can be zero?  If costs are strictly greater than zero then this would not happen?  (Well, even then, if you were really evil I suppose you could define an infinite number of actions with an infinite number of nonzero costs…)
p. 2-3: please say something about the complexity of the plan forbidding transformations.  Presumably they are polynomial time, but are they linear in the size of the set of plans?  Worse?
p. 3 proof of theorem 2: just to make sure I understand, it seems ‘certifying the transformed task $\Pi_P$’ means ‘find an optimal solution to $\Pi_P$ and make sure this solution has value > q’, which takes the usual amount of time for optimal planning, i.e. PSPACE or EXPTIME or what have you.  Right?
p.3: thm 3.  So you give me a propose solution to dominance planning.  But to check it I need to take this solution and extend it with all plans of top quality.  Do you give me \overline{P} and P?  Or do I have to compute \overline{P} by solving the top quality problem?   What is the complexity of this check?  (It’s dominated by the 2d bullet which requires me to solve a planning problem for every plan in P
p. 3: the forbidding plans with loops has to take a loop less plan to begin with for the forbidding translation, right?  This isn’t made explicit; suggest adding a sentence to this effect.

**Reproducibility:**

2: Some details are missing, but the paper still appears to be replicable with some effort.

**Strengths Of The Paper:**

The problem area  of Top-K is an interesting topic of current planning research.  The idea that many disparate problems can be unified and brought under a single banner is nice.  The result for loopless planning is also an interesting advance.

**Weaknesses Of The Paper:**

The paper is short and terse, which is unfortunate given that it also contains supplemental material.  Given that some of the material is complex, a longer paper with more explanations and more complete proofs would be more accessible.

There are no empirical results showing how long it takes to perform the certification operations.  While this is primarily a theory paper, results on complexity alone don't really give a sense of how 'hard' the certification problem is.

---

> ### Author Rebuttal · Authors · 2024-01-27
>
> We thank the reviewer for the feedback.
>
> * Our setting in this paper is classical sequential planning where plans are sequences of actions. But even in the case of “parallel” planning, assuming that partial order plans are specified as a relation over action instances, even if plans are partial orders instead of total orders, there can be multiple different partial orders over the same multi-set of actions and therefore there is a difference between unordered top-quality and top-quality planning. In the case of unordered top-quality these two will be equivalent, but in the case of top-quality these two will be different plans. In other words, not all equivalent plans can be encoded with a single partial order. A simple example is these four plans: abcd, bacd, cdab, cdba. These plans can be represented by two partial orders: {(a,c),(b,c),(c,d)} and {(c,d),(d,a),(d,b)}. There is no single partial order that can represent these 4 plans.
>
> * Yes, unordered top-quality solutions can be of infinite size due to 0-cost actions. In classical planning, we assume a finite set of actions.
>
> * The existing transformations are low-polynomial, but not linear in the input planning task size and the size of P. They increase the number of variables, the number of actions, and the size of each action (preconditions/effects) times the number of actions on plans in P.  While the new loopless top-quality transformation also increases the number of variables/actions times the size of P, it introduces disjunctive preconditions. When the disjunctive preconditions are compiled away, it can now again increase the number of actions quadratically. Note that a planning task size by itself is not necessarily a good predictor of planner behavior.
>
> * Indeed, the computationally extensive part is certifying optimality of the transformed task, which is as hard as optimal planning.
>
> * Theorem 3 (and 4): The input is indeed P, and \overline{P} must be computed from P and R. This part can be computationally challenging in some cases and even impossible in some cases as \overline{P} can be infinite for finite P (please see the loopless relation example in the paper). Even in cases like unordered top-quality, finding all orderings of the multi-set of actions that are plans can be computationally hard, but is in NP.
>
> * Thank you for the catch! If (\pi,\pi’)\in R_{ll}, then \pi must be loopless, but we did not mention it (albeit assumed) in theorem 5.

---

### Official Review · Reviewer_VmPT · 2024-01-24

**Significance And Importance:** 2
**Soundness:** 3
**Novelty:** 2
**Clarity:** 4
**Overall Evaluation:** 1
**Confidence:** 3

**Weaknesses:**

1: Minor weaknesses that are easily fixable.

**Contributions Of The Paper:**

The paper: (1) proposes a unified representation for different variations of top-quality planning problems using the generalized dominance relation; (2) provides certification for top-quality, top-k, and the newly introduced dominance top-quality plans; (3) provide a transformation from loop-less top-quality planning to top-quality planning by introducing new tasks and constraints to prevent loops.

**Ethical Considerations:**

(1) Not Applicable: The paper does not have any ethical considerations to address

**Nomination For Best Paper:**

No

**Questions For Authors:**

1. Did you try to implement the algorithms presented in this paper: (1) to certify top-quality, dominate top-quality, top-k planning combining optimality certification + forbidding plan transformation; (2) transformation to forbid plans with loop and if so are they efficient to solve?

**Reproducibility:**

0: N/A - nothing to reproduce.

**Strengths Of The Paper:**

I agree with the authors that there are too many variations of top-quality/top-k planning problems (ordered, unordered, subset, loop-less, partial ordered etc), and that there will be more to come, and they are currently only defined for classical planning so far, which do not have many real-world applications. As we get into more practical planning setting with time, numeric variables, resources, uncertainty, task-network etc, there will be even more variations that can be proposed. Thus, some unified “dominance” representation, at least covering existing variation is welcomed.

The utilization of existing method to certify optimal planning and plan-forbidding transformation to provide certificate for top-quality and top dominance quality are clever and good contributions.

The paper is well-written.

**Weaknesses Of The Paper:**

I'm not clear about the overall impact of the paper for future development in the "top-quality" planning theme. Researchers will come up with new variations and propose approaches specifically designed to solve such variations. While those variations will likely fall into this "dominate top-quality" umbrella, I'm not sure yet if ideas and techniques proposed here will influence or get used in such work.

While I understand that probably each domination relation will warrant different techniques best suited to solve them and it’s ok to not presenting empirical evaluation for dominance top-quality planning. However, some empirical evaluation for the transformation presented in the “Efficient Computation” would be good to show that it’s actually efficient and practical. Since the transformation introduce new preconditions and effects and copies of the existing tasks, it’s not clear if solving the transformed problem is more efficient than utilizing the existing approach so it would be nice to back it up empirically.

---

> ### Author Rebuttal · Authors · 2024-01-27
>
> We thank the reviewer for the feedback.
>
> The certification of the set of plans P as dominance top-quality planning solution takes P and the planning task \Pi as input and returns true or false. The transformations mentioned in the section “Efficient Computation” for unordered top-quality and subset top-quality are polynomial in the size of the input. An implementation of these transformations already exists in the forbiditerative planner (Katz & Sohrabi 2020, 2022). It takes milliseconds to transform a task and is done at every iteration of the aforementioned planner.  The new proposed transformation for loopless top-quality planning is polynomial in the size of the planning task, the number of plans in P, but also in the lengths of the plans in P. It is typically larger than the previously mentioned transformations, but is also not expected to take a significant amount of time.
>
> The certification of optimality of the resulted planning task is the time consuming part of the process. It is computationally hard (Mugdan et al 2023) and can be done by, e.g., a certified cost-optimal planner. Experimental results showing how long does it take to certify optimality on IPC benchmarks can be found in  Mugdan et al (2023).
>
> The certification procedure for top-quality planning variants would then only use existing components sequentially.
>
> As our work only presents one efficient method per top-quality variant, we do not have a frame of reference for additional experiments.

---

### Meta-Review · Area_Chair_8msb · 2024-02-04

**Recommendation:** Accept (Poster)
**Confidence:** 2

**Metareview:**

This paper is sort of borderline. There is nothing inherently wrong with it, but all reviewers agree that a longer paper, containing more background on the different variants of the top-quality problem, the compilations used (including their complexity, and an empirical evaluation of them in comparison with other methods) would be better. (If the aim is to unify a set of different problem formulations, then it makes sense to also present a thorough survey of work on those problems.) The question comes down to how valuable it will be to the planning community to have this short paper published first (hopefully to be followed by a longer one that includes all these things) vs having only a longer paper published in the (hopefully near) future. The majority (2 out of 3) arrived at the answer that this short paper would have some value, and hence the recommendation is to accept, but it is a weak recommendation.

**Ethical Considerations:**

(1) Not Applicable: The paper does not have any ethical considerations to address